# The Molecular Mechanism of Yam Polysaccharide Protected H_2_O_2_-Induced Oxidative Damage in IEC-6 Cells

**DOI:** 10.3390/foods12020262

**Published:** 2023-01-06

**Authors:** Mingyue Shen, Ruixin Cai, Zhedong Li, Xiaodie Chen, Jianhua Xie

**Affiliations:** State Key Laboratory of Food Science and Technology, Nanchang University, Nanchang 330047, China

**Keywords:** intestinal barrier, yam polysaccharide, RNA-seq, oxidative damage, signaling pathways

## Abstract

Oxidative stress is involved in maintaining homeostasis of the body, and an in-depth study of its mechanism of action is beneficial for the prevention of chronic illnesses. This study aimed to investigate the protective mechanism of yam polysaccharide (CYP) against H_2_O_2_-induced oxidative damage by an RNA-seq technique. The expression of genes and the function of the genome in the process of oxidative damage by H_2_O_2_ in IEC-6 cells were explored through transcriptomic analysis. The results illustrated that H_2_O_2_ damaged cells by promoting cell differentiation and affecting tight junction proteins, and CYP could achieve cell protection via restraining the activation of the MAPK signaling pathway. RNA-seq analysis revealed that H_2_O_2_ may damage cells by promoting the IL-17 signaling pathway and the MAPK signaling pathway and so forth. The Western blot showed that the pretreatment of CYP could restrain the activation of the MAPK signaling pathway. In summary, this study demonstrates that the efficacy of CYP in modulating the MAPK signaling pathway against excessive oxidative stress, with a corresponding preventive role against injury to the intestinal barrier. It provides a new perspective for the understanding of the preventive role of CYP on intestinal damage. These findings suggest that CYP could be used as oxidation protectant and may have potential application prospects in the food and pharmaceutical industries.

## 1. Introduction

Oxidative stress is a response of the body’s cells and tissues to stimuli that increase reactive oxygen species and free radicals, resulting in the dysregulation of the intracellular oxidative–oxidative system [1]. When oxidative stress occurs in cells, the expressions of superoxide dismutase, catalase, and glutathione peroxidase in the body are significantly altered. Meanwhile, the harmful superoxide anion generated is catalyzed to H_2_O_2_, which is further decomposed into water and molecular oxygen [2,3]. Oxidative stress also disrupts the intestinal environmental balance and changes intestinal permeability, giving rise to intestinal damage and triggering a range of intestinal diseases [4]. Stress response can induce intestinal cells to produce ROS metabolites in quantity, lipid peroxidation, protein denaturation, cell apoptosis, and, ultimately, resulting in intestinal mucosal damage and inflammatory intestinal diseases [5].

Irritation of the gastrointestinal tract by multiple factors can lead to gastrointestinal dysfunction, which is one of the most vulnerable systems [6]. The intestinal barrier is considered to be the first line of defense against host invasion by pathogens [7]. Imbalances in the gut microbiome and impaired intestinal mucosal barrier have been linked to damage from inflammation, oxidative stress, and various systemic diseases, such as inflammatory bowel disease (IBD), irritable bowel syndrome (IBS), liver fibrosis, diabetic nephropathy, lupus nephritis, and sepsis [8,9]. H_2_O_2_ is naturally present in various cells of the body and also stimulates cells by oxidative stress that occurs with the body, and excess H_2_O_2_ will disrupt the oxidative–antioxidant balance within the cells, giving rise to the production of ROS [10], altering the permeability of intestinal cells, as well as decreasing the intestinal barrier function [9]. There are many studies investigating the activation of related proteins on intracellular signaling pathways via Western blot technique, but there are few reports on the expression of intracellular tight junction proteins in intestinal epithelial cells in response to some stimuli, and the intracellular signaling pathway activation pathways are still not clear.

Polysaccharides extracted from yams by water extraction and alcohol precipitation methods have biological effects, such as antitumor, antioxidant, and modulating immune activity [11,12]. In vitro antioxidant activity of floral mushroom polysaccharides showed that they scavenged 79.46% and 74.18% of DPPH and hydroxyl radicals (OH), respectively [13]. The effect of yam polysaccharides on I-type diabetic mice has also been reported, and after treatment with yam polysaccharides, the reactive oxygen species and malondialdehyde (MDA) content in diabetic mice were reduced, indicating that yam polysaccharides have antioxidant effects in diabetic mice, thus validly guarding against multifarious complications of diabetes [14]. In addition, yam polysaccharides can reduce DPPH radicals, hydroxyl radicals, and superoxide radicals’ production so as to promote endometrial epithelial cell proliferation [15,16]. However, the recent studies on the antioxidant activity of yam polysaccharides mainly focused on the scavenging ability of free radicals in vitro, while the role of oxidative damage at the cellular level is still less and the mechanism of action has not been reported. Therefore, to explore the protective effect of yam polysaccharides against oxidative damage from the cellular level is need.

RNA-Sequencing (RNA-seq) is regarded as a new method to analyze gene functions and interactions at the histological level, and the current RNA-seq method can determine the expression levels of all most genes, which has been broadly used in frontier fields, such as molecular biology [17,18]. Wang et al. [19] analyzed the resistance of periplaneta Americana peptide to human ovarian granulosa cell apoptosis under hydrogen peroxide poisoning, and RNA-seq analysis showed that activation of IL-6 trans-signaling pathway in retinal endothelial cells caused gene expression changes. In addition, the combined RNA-seq and cytobiology techniques illustrated that Cyclocarya paliurus polysaccharide has protective effects against H_2_O_2_-induced oxidative damage in L02 cells and regulates mitochondrial function, oxidative stress, and PI3K/Akt and MAPK signaling pathways [20].

This study aimed to probe the underlying molecular mechanism of CYP against H_2_O_2_-induced oxidative lesions and the protective role on the gut barrier by transcriptomic sequencing technology based on the IEC-6 cell model. 

## 2. Materials and Methods

### 2.1. Reagents

Fresh yams (Dioscoreae Rhizoma) were purchased from Jiujiang, Jiangxi, China. Fetal bovine serum (FBS) was obtained from Viva cell Biosciences Ltd. (Shanghai, China). Dulbecco’s modified eagle medium (DMEM) was purchased from Solarbio (Beijing, China). Antibodies were gained from Cell Signaling Technology (USA). The IEC-6 cell line was purchased from the Cell Bank of the Chinese Academy of Sciences (Shanghai, China).

### 2.2. Preparation of CYP 

We weighed 300 g of yam powder and added 6 L of ultrapure water in the ratio of 1:20. Then, we stirred while heating at 80 °C, centrifuged, and concentrated the mixture to 1/10 of the original volume. Then, we added 95% ethanol and precipitated the mixture 4 °C overnight. The next day, the precipitate was redissolved and treated with glycosylase, α-amylase, and papain after adjusting the pH of the solution to 4.5 and 6.5, respectively. Finally, the Sevage method was used to deproteinization, followed by dialysis in a dialysis bag (retention capacity of 8000–14,000 Da). The dialyzed solution was lyophilized after secondary alcohol precipitation with anhydrous ethanol to obtain the crude polysaccharide [21,22]. The contents of carbohydrate, uronic acid, and protein were 33.62%, 34.95%, and 5.26%, respectively, and the molecular weight was 20.89 kDa [23].

### 2.3. Cell Culture

IEC-6 cells in good growth condition were taken and inoculated in 6-well plates at a cell density of 2 × 10^5^ cells/mL for subsequent experiments. Cells were pretreated with different concentrations of CYP for 24 h after cell apposition, and the polysaccharide samples were incubated with the configured H_2_O_2_ solution for 4 h. Groups were divided as follows: Blank control group—cells were cultured daily with medium containing 10% FBS; Model group—cells were cultured with medium containing 10% FBS and treated with 300 μmol/L H_2_O_2_ solution on the last day; Low-concentration polysaccharide group—treated with 200 μg/mL of yam polysaccharide solution for 24 h, followed by 300 μmol/L of H_2_O_2_ solution; Medium-concentration polysaccharide group—treated with 400 μg/mL of yam polysaccharide solution for 24 h, followed by 300 μmol/L of H_2_O_2_ solution; and High-concentration polysaccharide group—treated with 800 μg/mL of yam polysaccharide solution for 24 h, followed by 300 μmol/L of H_2_O_2_ solution.

### 2.4. RNA-Seq Analysis

Total RNA was collected and purified from cell samples using the Trizol reagent kit (Invitrogen, Carlsbad, CA, USA) on the grounds of the manufacturer’s certificate, and subsequent library preparation and sequencing was performed at Shanghai Personal Biotechnology (Shanghai, China). RNA-seq libraries were identified using an Agilent 2100 Bioanalyzer (Agilent Technologies, Palo Alto, CA, USA) to evaluate the quality of the RNA. Appropriate libraries were sequenced on the Illumina NextSeq500 platform (Illumina, San Diego, CA, USA).

Read Count values were statistically compared to each gene using HTSeq as the benchmark for gene expression. DESeq was used to analyze the difference of gene expression, and |log2 fold change| > 1 and *p* < 0.05 were selected as the condition of screening differentially expressed genes (DEGs). In Gene Ontology (GO) analysis, topGO was used to map the DEGs to their specific directed acyclic graphs (DAGs) structure, and the intensity of the color depended on the enrichment fraction. Then, the *P*-value was estimated by a super geometric allocation method to calculate the *p*-value (significant enrichment is defined as *p* < 0.05) to recognize GO terms that were apparently enriched in differential genes in comparison with the entire genomic background, and thus determined the primary biological effect exercised by the differential genes. KEGG enrichment analysis was executed with the cluster-profiler where the gene list and gene number of each pathway were estimated through the differential genes annotated in the KEGG pathway, and then the *P*-value was calculated by the hypergeometric allocation method (the criterion for significant enrichment was *p* < 0.05) to pick out the target genes. The KEGG pathways that are significantly enriched in differential genes contrast with the gross genomic background are then used to identify the major biological functions executed via the differential genes.

### 2.5. Inhibition of MAPKs Using Specific Inhibitors 

PD98059 is a potent and selective ERK inhibitor that specifically inhibits MAPK kinase activation in both in vivo and in vitro experiments [24,25]. In order to judge if the activating effect of CYP on IEC-6 cells is via the MAPK signaling pathway, the PD98059 inhibitor was added to DMEM medium and configured as a 10 μM solution. The configured ERK inhibitor solution was added an hour prior to the addition of H_2_O_2_ damage to the cells. The proteins in the cells were extracted at the end of H_2_O_2_ damage and Western blot was applied to estimate the expression of the relevant proteins.

### 2.6. Western Blot Analysis

Western and IP Cell Lysis Solution were used for lysing cells at −20 °C and then the lysate was transferred to an EP tube and centrifuged (4 °C, 10,000 r/min, 8 min.) After centrifugation, the supernatant was collected and the protein concentration was estimated with the BCA kit in line with the procedure, followed by the addition of SDS-PAGE loading buffer (6×) and denaturation at 95 °C for 5 min. After the electrophoresis, the gel was cut out at the destination position according to the molecular weight, indicated by the marker. The electrode was covered and sent to the membrane transfer apparatus to transfer the membrane at 25 V for 7 min. After the transfer, the protein strips were cut according to the experimental purpose, submerged in the blocking solution (5% BSA), and shaken at room temperature for 1 h. The target primary antibody was selected for overnight incubation, and the secondary antibody was added after three washes with TBST for 15 min each time on the next day. Eventually, target protein bands were captured and photographed by chemiluminescence imaging system.

### 2.7. Statistical Analysis 

All experimental values are from at least three tinning sessions and data were expressed as mean ± standard deviation (SD). Differences among data mean values were tested for statistical significance at the *p* < 0.05 level using One-way ANOVA analysis of variance by SPSS 17.0 (SPSS Inc., Chicago, IL, USA).

## 3. Results and Discussion

### 3.1. Purity and Quality of Isolated RNA

The quality of the resulting RNA was evaluated by agarose gel electrophoresis. The total RNA electrophoresis bands of the samples were clear, with a total of three bands, from large to small, 28S, 18S, and 5S, with decreasing brightness. The 28S band was brighter than the 18S band (Figure 1), which was twice as bright, while 5S was very light, in normal condition, without obvious trailing and smears, with good RNA integrity and no significant degradation of mRNA. The results mentioned met the requirements of the subsequent experiments and could be carried out in the next step.

### 3.2. Expression Difference Result Statistics

RNA-seq is broadly applied to study molecular mechanisms associated with the pharmacological activity of natural drugs’ polysaccharides [26,27]. Figure 2A shows the differential gene values in the form of bar graphs. In the blank control group compared with the CYP group, it identified 2637 gene sequences and the up-regulated genes consisted of 1563 and the down-regulated ones were 1074; in the model group compared with the CYP group, it identified 1407 differentially expressed genes, of which the up-regulated genes consisted of 358 and the down-regulated ones were 1049; in comparison to blank and model groups, there were 3712 differential gene sequences, of which 2630 were upwardly adjusted and 1082 were downgraded. The heat map shown in Figure 2B is expressed at the gene level, with one sample in each column, with red for highly expressed genes and blue for low-expressed genes. Due to the similarity of gene expression mode, gene sequence samples can be clustered for analysis. It can be seen that the correlation coefficients are greater than 0.9 for samples within the same group and less than 0.8 for samples between different groups (Figure 2C), indicating good sample parallelism between groups. On the basis of results of the one-way analysis of variance (ANOVA), the Wayne plot was able to calculate the number of unique differential genes shared between the comparable groups. The number of genes differing among the comparison groups and the overlapping relationship among the comparison groups were indicated. In the Venn diagram (Figure 2D), it can be seen that there are the same 417 differentially expressed genes in the three two-by-two comparisons.

### 3.3. Functional Prediction Analysis by GO Enrichment 

GO enrichment analysis starts by mapping all DEGs into each term of the GO database, using the whole genome as the background, subsequently calculating the *P*-value by hypergeometric distribution algorithm to determine the GO term, which is prominently captured in differential genes within the whole genome background, thereby ascertaining the biological effect represented by the differential genes [28]. The regulatory role of CYP on IEC-6 cells is still unclear at the transcriptome level. Figure 3B–D showed the classification of biological processes (BP), molecular functions (MF), and cellular components (CC) on the grounds of sequencing analysis, respectively. GO function analysis revealed 26 BP terms, 20 MF terms, and 18 CC terms in the DEGs.

The functional groups that changed significantly in terms of model group and control group were the development of anatomical structures, course of development, development of multicellular organisms, cell differentiation, processes of cell development, system development, positive regulation of biological processes, and protein binding. The functional groups that changed significantly in the CYP group were the development of multicellular organisms, system development, development of anatomical structures, course of development, regulatory signals, regulation of cell communication, the developmental processes, cell differentiation, development of animal organs, regulation of responses to stimuli, regulation of signal transduction, negative regulation of biochemical reactions, movement in motor cells or subcellular components, protein binding, positive regulation of biochemical reactions, signal transduction regulation, and cell differentiation. The functional groups that changed remarkably in the CYP group compared to the model group were the development of anatomical structures, cell developmental processes, system development, development of multicellular organisms, cell differentiation, regulation of multicellular biological processes, bio-adhesion, cell adhesion, cell surface receptor signaling pathways, regulation of cell differentiation, regulation of multicellular biological development, and metal ion transport activity across membranes (Figure 3A). The above sequencing data indicates that CYP can regulate the mechanism of oxidative damage in IEC-6 cells.

### 3.4. Signaling Pathway Analysis by KEGG Enrichment 

The KEGG database integrates genomic, biochemical information, and signal pathways condition in one, using the pathway mapping process to link the genome to life as a reference [19]. To identify H_2_O_2_-induced signaling pathways activated by IEC-6 cells, the quantity of DEGs in the KEGG analysis were calculated. KEGG analysis was confirmed with bubble plots showing the top 20 canonical pathways.

The pathways that changed significantly in the model group compared to the control group were IL-17 signaling pathway, tumor necrosis factor, MAPK, and cAMP signaling pathways, cytokine receptor interactions, endocrine resistance and atherosclerosis, TGF-beta pathway, and FoxO. The pathways that changed more significantly in CYP were tumor necrosis factor, neuro glioma, MAPK, p53, PI3K-Akt, IL-17, cytokine–cytokine receptor interaction, HIF-1, FoxO, calcium signaling pathways, cellular senescence, and cell cycle. The functional groups that changed remarkably in the CYP group in comparison to the model group were tumor necrosis factor signaling pathway, IL-17 signaling pathway, protein digestion and absorption, cytokine–cytokine receptor interaction, cGMP-PKG signaling pathway, and mineral uptake. The MAPK pathway was significantly changed in the control group compared with both the model group and CYP group (Figure 4). Therefore, the above results indicated that CYP may make sense in protecting cells from oxidative damage by affecting MAPK signaling pathway subsequent validation from the MAPK pathway as proposed.

### 3.5. Effect of ERK Inhibitor on MAPK Pathway of H_2_O_2_-Induced Oxidative Damage in IEC-6 Cells

Our preceding study has illustrated that the phosphorylated expression of three proteins, JNK, ERK, and P38, in the MAPK pathway activated by H_2_O_2_ was significantly reduced in the presence of CYP, which was able to inhibit the activation of the MAPK pathway [23]. In consequence, to bear out that activating the MAPK signaling pathway in RNA-seq results, ERK inhibitor (PD98059) was added before treating the cells with H_2_O_2_. Compared with the H_2_O_2_ group, pretreatment with ERK inhibitor was effective in reducing the ratios of p-ERK to ERK, p-P38 to P38, and p-JNK to JNK, and the phosphorylation content of both JNK and ERK proteins’ gray value was declined by the co-treatment of ERK inhibitor with CYP, but not as strongly as CYP alone (Figure 5). Our results indicated that the combination of ERK inhibitor with CYP did not have an enhanced inhibitory effect, which may be due to the negative effect of the addition of PD98059 on the protective effect of CYP. According to the above results, CYP has a similar effect to ERK inhibitors in declining the expression of protein phosphorylation and inhibiting the activation of MAPK pathway.

### 3.6. Effect of CYP on Intestinal Tight Junction Proteins ZO-1, Occludin, and Claudin-1

Intestinal tight junction proteins have a positive impact on protecting intestinal cells as a barrier for intestinal mucosal cells [29]. The, ZO-1, Claudin-1, and Occludin expressions were decreased after H_2_O_2_ treatment compared with the control group (Figure 6), indicating that H_2_O_2_ treatment impaired the intestinal barrier function. After pretreatment with CYP at 200 μg/mL, the expression of all three tight junction proteins was up-regulated to some extent, whereas when the concentration of CYP was increased to 400 μg/mL, the expression of Occludin showed a decrease, and when soaring again to 800 μg/mL, the expression content of ZO-1 and Claudin-1 was lower than that of the control group. The results illustrated that the gut mucosal barrier of IEC-6 cells can be protected under low concentration of CYP treatment, but too-high concentration of CYP can have a negative effect.

## 4. Discussion

The protective mechanism of yam polysaccharides in opposition to H_2_O_2_-induced oxidative lesions in IEC-6 cells was investigated based on RNA-seq technology, while the protective efficacy of CYP against H_2_O_2_-induced injury in IEC-6 cells was explored in the MAPK pathway, and the effect of combined use of ERK inhibitors on MAPK pathway was further selected to validate the effect of CYP based on the results of KEGG enrichment analysis.

RNA sequencing is a method that allows the evaluation of a complete set of organisms’ transcribed genes, non-coding RNAs, and their transcriptomes [27]. Therefore, we used sequencing to screen all transcripts of CYP against H_2_O_2_-induced oxidative damage in IEC-6 cells. In this study, the expression of differential genes was detected via using RNA-seq technology and further classified based on the information from the differential gene analysis to analyze which functions these genes are referred to. Experimental results illustrated that there are many differentially expressed genes in both CYP and model groups versus the control group, manifesting a large number of genes expressed after H_2_O_2_ stimulation of cells with pretreatment of CYP. GO database provides biological annotation of sample genes in the light of effect, biological pathway, and cellular localization [19,30]. GO enrichment analysis showed that the functional groups were significantly changed during this process with cell differentiation and development of multicellular organisms and so on, indicating that CYP can influence the physiological function of IEC-6 cells. In addition, in comparison to GO analysis, KEGG enrichment analysis possesses a mighty graphical representation, which could utilize diagrammatic form instead of text material to demonstrate a range of signal pathways, offering more intuitionistic and all-sided information. For the sake of investigating the signaling pathway of CYP affecting IEC-6 cells, we found that MAPK and IL-17 signaling pathways were remarkably enriched. 

Enzymes in the MAPK signaling pathway jointly regulate cell growth, differentiation, environmental stress adaptation, inflammatory response, and other important cell physiology [31]. The functional groups that are significantly changed from inside the GO enrichment analysis are cell differentiation, which coincides with the function of the MAPK pathway. The MAPK signaling pathway is mainly activated through the sequence of MAPK kinase, phosphorylation-activated MAPK kinase, and MAPK (transmission of downstream information) activation to complete [25]. To keep off mutual interference, the MAPK cascade is split into fixed modules using scaffold proteins. Among mammals, the MAPK signaling cascade is usually classified into four subgroups: the extracellular signal-regulated protein kinase 1 and 2 (ERK1/2) cascade, the c-Jun N-terminal kinase (JNK) cascade, the p38 MAPK cascade, and the ERK5 cascade [32,33]. These subgroups are connected to signaling pathways with various biochemical actions. For example, ERK1/2 gains command of cell growth and differentiation [34], JNK is correlated with stress, p38 MAPK is relevant to immune regulation and inflammatory response [35], and ERK5 is associated with multifarious diseases, especially cardiovascular disease and cancer [36].

Western blot protein bands indicated that pretreatment with CYP inhibited the phosphorylated expression of three proteins in the pathway and suppressed MAPK pathway activation. Further validation with the combination of ERK inhibitors showed that CYP exerted an inhibitory effect on the phosphorylated expression of the related proteins in the MAPK pathway and that the effect was similar to that of ERK inhibitors. This indicates that CYP can inhibit the MAPK pathway and, thus, reduce the oxidative damage in cells. Consistent with our results, polysaccharides from Agrocybe cylindracea residue mitigate the liver and colon injury in type II diabetes through p38 MAPK signaling pathway [37]. Maintaining the integrality of the intestinal mucosal barrier is essential to protecting the host from gut microbes, food antigens, and toxins. Huang et al. [38] investigated the effects of CYP and SCYP on intestinal microbiota in vitro study. Consequently, it is speculated that CYP may affect the gut barrier. The present study found that low concentration of CYP can promote the expression of intestinal compact junction protein, which plays a role in protecting the intestinal tract.

## 5. Conclusions

To sum up, our study demonstrated that cells differentiated and protein binding genes changed significantly according to the sequencing results and CYP-regulated MAPK signaling pathways with ERK inhibitor to prevent excessive oxidative stress responses and protect the intestinal barrier by affecting intestinal proteins (Figure 7). Indeed, this is the first comprehensive study of the potential protective mechanism of CYP, which provides an original perspective for incisive understanding of the protective role of CYP on the gut, and likewise offers a theoretical foundation for the CYP-relevant products’ development.

## Figures and Tables

**Figure 1 foods-12-00262-f001:**
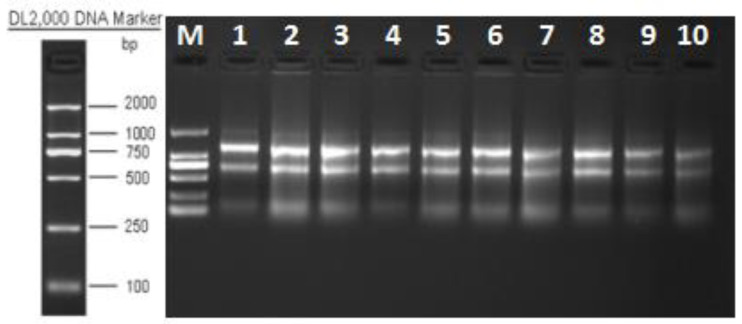
Determination of total RNA quality.

**Figure 2 foods-12-00262-f002:**
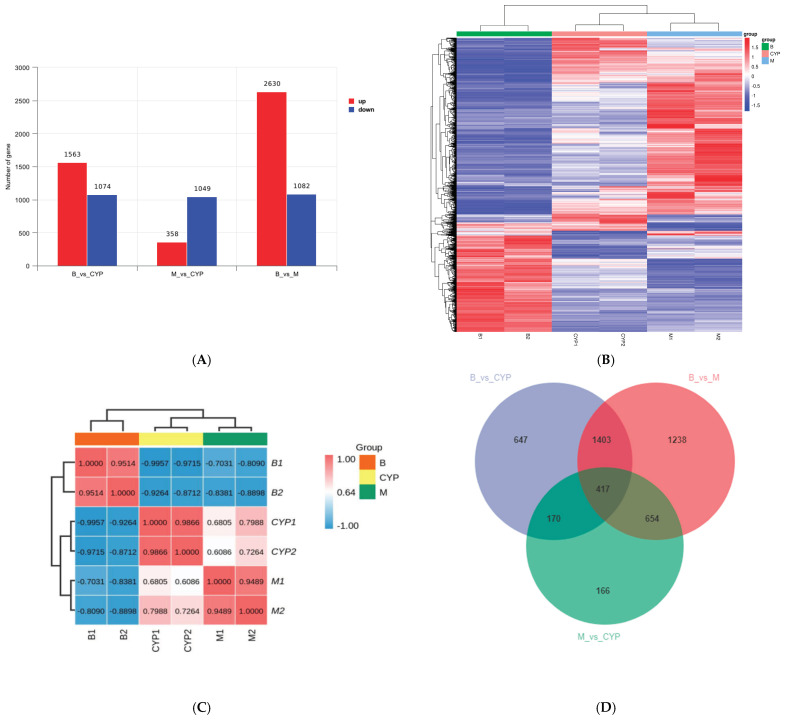
Changes in the mRNA expression profile of IEC-6 cells induced by H_2_O_2_ after CYP pretreatment. (**A**) Statistics of differentially expressed genes between groups. (**B**) DEGs thermogram representation between groups. (**C**) Correlation test between groups. (**D**) Venn diagram between groups.

**Figure 3 foods-12-00262-f003:**
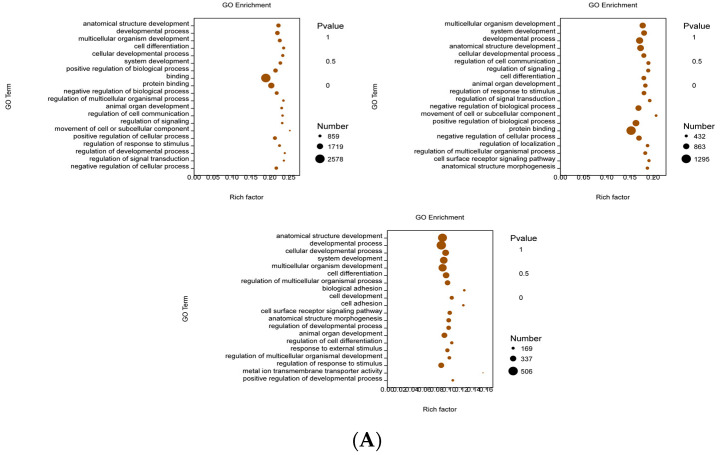
GO enrichment analyses of DEGs identified in H_2_O_2_-stimulated IEC-6 cells. (**A**) GO enrichment analyses of DEGs. The *x*-axis represents the number of DEGs and the *y*-axis represents the enriched GO terms. The size of the point indicates the number of DEGs enriched in the pathway, and the redder the color, the more significant the enrichment result is. (**B**–**D**) Thumbnails view of directed acyclic graphs (DEGs) on BP, CC, and MF.

**Figure 4 foods-12-00262-f004:**
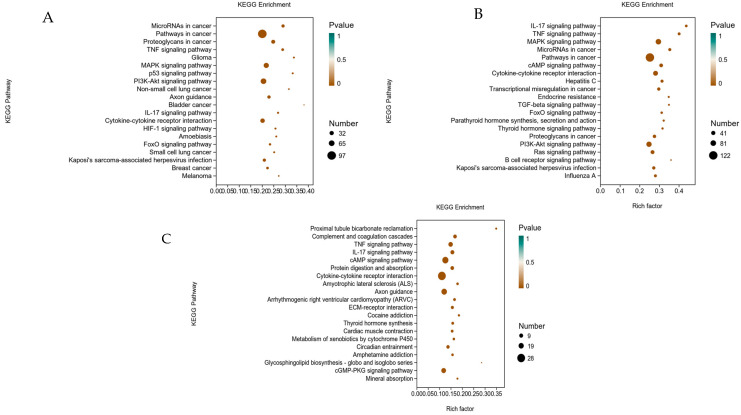
Enrichment analysis diagram of KEGG in H_2_O_2_-stimulated IEC-6 cells. (**A**) Control group, (**B**) Model group. (**C**) CYP group. The top 20 classical pathways obtained through KEGG enrichment are shown in the figure.

**Figure 5 foods-12-00262-f005:**
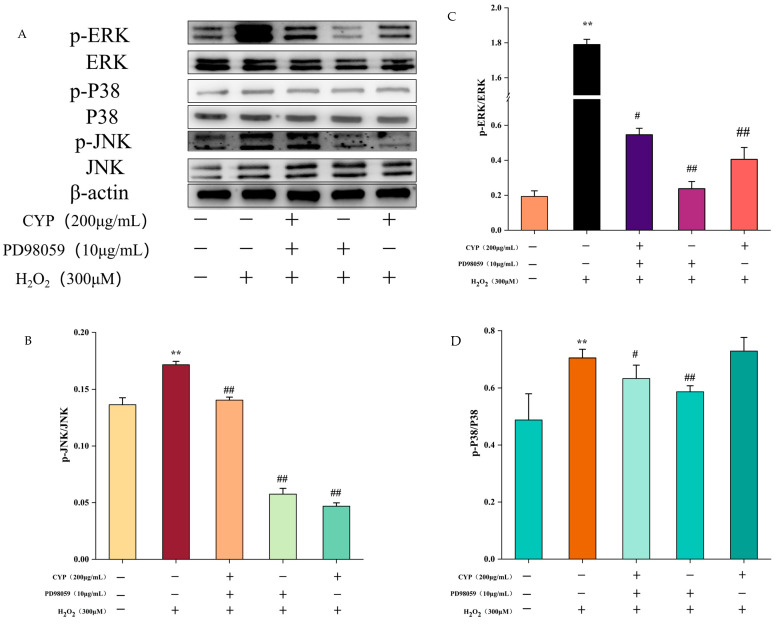
Effect of ERK inhibitor on H_2_O_2_-induced IEC-6 cell injury through MAPK pathway. (**A**) Western blot showing the protein expression of ERK, p-ERK, p38, p-p38, JNK and p-JNK. (**B**–**D**) The quantification of JNK, p-JNK, ERK, p-ERK, p38, and p-p38 expression. Results shown are expressed as means ± SD (n = 3). *# p* < 0.05, *## p* < 0.01 compared with normal group, *** p* < 0.01 compared with H_2_O_2_ group alone.

**Figure 6 foods-12-00262-f006:**
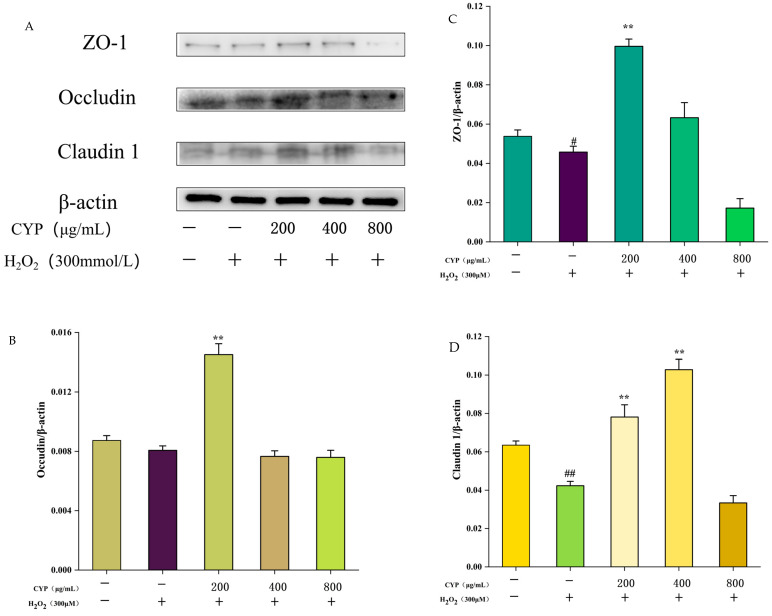
Effects of CYP on ZO-1, Occludin, and Claudin-1 levels of H_2_O_2_-induced IEC-6 cells. (**A**) Western blot showing the protein expression of ZO-1, Occludin and Claudin-1 in IEC-6. (**B**–**D**) The quantification of Occludin, ZO-1, and Claudin-1 expression. Results shown are expressed as means ± SD (n = 3). *# p* < 0.05, *## p* < 0.01 compared with normal group, *** p* < 0.01 compared with H_2_O_2_ group alone.

**Figure 7 foods-12-00262-f007:**
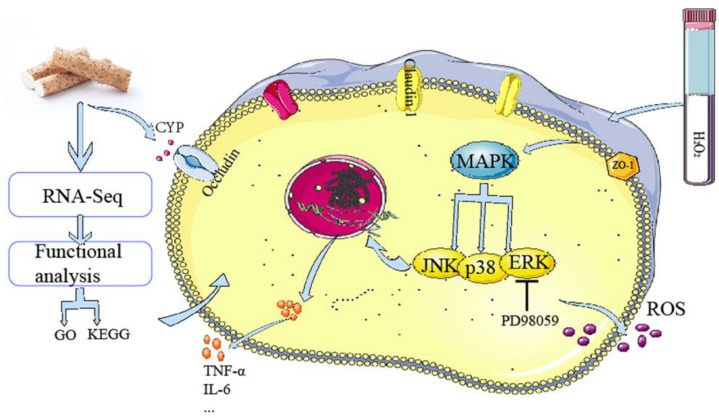
Schematic of mechanism in CYP on H_2_O_2_-induced oxidative damage in IEC-6 cell.

## Data Availability

The data used to support the findings of this study can be made available by the corresponding author upon request.

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
