# Peer review of "The Molecular Mechanism of Yam Polysaccharide Protected H_2_O_2_-Induced Oxidative Damage in IEC-6 Cells"

_foods, 2023, doi:10.3390/foods12020262_

Round 1

Reviewer 1 Report

The manuscript entitles “RNA-seq analysis identifies the molecular mechanism of yam polysaccharide protection against H2O2-induced oxidative damage in IEC-6 cells through MAPK signaling pathway” has been written well and have some comments below:

Title is looking very big, so it should be concise.

In abstract please put final recommendation of this study in the end.

-In keywords, please put keywords different from the title.

-In methodology part, 2.1, no need to give full details so in my opinion it may be removed.

Line no 123, Differential expression genes were conform to the fold |log2FoldChange|>1 and significance P <0.05. What do you mean from this sentence? Please correct and rewrite.

Line no 125, correct the sentence, GO enrichment analysis was carried out with topGO to calculate gene lists and number of genes….” Also give full form of “GO” before using first time in the text.

- Figure quality is not so good, so please upload high resolution figures as it is not seen what do you want to show there.

-Figure 7, should put in appropriate place of result or methodology portion from the conclusion.

-Which test you used for statistical analysis?

-      In Conclusion: don’t use figure. Put final concluding recommendation and outcome of this study.

-       There are so many References, in which you didn’t addressed fully so remove this anamoly and put full reference as per journals instruction.

English language should be improved.

Reviewer 2 Report

Comments to the Author
Authors should address the following questions.
1. Section 2.3: Lack of references to support the author's statement : add the rference of use of the concentration of the H202 and the polysacharide
2.  I would recommend to authors make some Analysis and structure, physico-chemical
a  of CYP to verify the   protection against H2O2-induced oxidative damage in IEC.

3. The figures 2 and 3  are not clear,

4. The reference : 21 : « Study on deproteinization in extraction of polysaccharides from Patentillaunserina by Sevage »I would recommend to authors to add  the number of the pages.
